# Balanced sampling for an object detection problem - application to fetal anatomies detection

**Antoine Olivier**[1]               ANTOINE.OLIVIER@PHILIPS.COM
and **Caroline Raynaud**[1]            CAROLINE.RAYNAUD@PHILIPS.COM
[1] *Philips Research, France*

## Abstract

In this paper, we propose a novel approach to overcome the problem of imbalanced datasets for object detection tasks, when the distribution is not uniform over all classes. The general idea is to compute a probability vector, encoding the probability for each image to be fed to the network during the training phase. This probability vector is computed by solving a quadratic optimization problem and ensures that all classes are seen with similar frequency. We apply this method to a fetal anatomies detection problem, and conduct a statistical analysis of the resulting performance to show that it performs significantly better than two baseline models: one with images sampled uniformly and one implementing oversampling.
**Keywords:** Object detection, Imbalanced dataset, Quadratic optimization, Fetal anatomy detection, Ultrasound.

## 1. Introduction and problem statement

In this paper, we tackle the issue of imbalanced datasets in object detection problems. Object detection consists in simultaneous identification and localization of objects in an image. Current state-of-the-art methods for object detection rely on deep learning algorithms, which require to train a model on a given dataset. For example, let us mention the classical two-stage networks Fast R-CNN and Faster-RCNN (Girshick, 2015; Ren et al., 2015), as well as the one-stage architectures SSD (Liu et al., 2016; Fu et al., 2017) and YOLO (Redmon et al., 2016; Redmon and Farhadi, 2016). However, balance between different classes is often hard to achieve, in particular in the field of medical imaging where data and annotations are costly and difficult to obtain.

**Imbalanced dataset for object detection.** An imbalanced dataset may affect the performance of a neural network, leading to poorer behavior of a model on under-represented classes. A common approach, consisting in oversampling all images containing the less frequent classes, could introduce unexpected and unwanted behavior, as it may also oversample examples of the most represented classes if they are jointly present in the images, and therefore not improve the overall performance of a model. We will elaborate more on this issue in Section 3.1.

The general issues with imbalance in object detection have been covered in the survey paper (Oksuz et al., 2019), dividing them into *class imbalance*, *scale imbalance*, *spatial imbalance* and *objective imbalance*. The problem of interest in this paper falls into the category of *class imbalance*, and more precisely in the so-called *foreground-foreground class imbalance*. Although objects appear at different frequencies in nature, and therefore class

imbalance is most likely to happen, it is stated in the survey (Oksuz et al., 2019) that 'imbalance amongst the foreground classes has not attracted as much interest as foreground-background imbalance', for which the works (Shrivastava et al., 2016; Lin et al., 2017; Pang et al., 2019; Li et al., 2019; Chen et al., 2019) can be cited.

For foreground-foreground imbalance, let us mention the papers (Ouyang et al., 2016; Oksuz et al., 2020). In (Ouyang et al., 2016) the authors investigate fine-tuning of a model on a dataset with *long-tail distribution* and 200 classes. They show that it is better to have (pseudo-)uniform number of samples per class, but sampling is done at bounding box level, before training the detector to classify each region, which is not easily generalizable to one-stage detectors. In (Oksuz et al., 2020) the authors present an online foreground balancing (OFB) method, aiming at making the classes balanced in a batch. Although their approach would apply to the same type of problems than ours, we point out that it is more suitable for two-stage detectors, as they generate positive bounding boxes after the region proposal network, whereas ours is agnostic to the network's architecture. The experiments we will present are conducted using the YOLO model, a one-stage detector. Besides, OFB makes the classes balanced at batch level, whereas our approach plays a role at dataset level. Note that it is a noticeable difference as, especially for small batch sizes, distribution within a batch may be different from distribution within the full dataset. Oksuz et al. (2019) lists as an open question whether OFB might induce a bias in the learning process.

Generative methods (Goodfellow et al., 2014) can also be used to produce artificial images (see (Tripathi et al., 2019; Wang et al., 2019)) for which special attention can be given to under-represented classes. One other noticeable approach is the one described in (Dwibedi et al., 2017), where object instances are simply 'cut' and then 'paste' on random backgrounds. Of course, this can result in unrealistic images.

We emphasize that the aforementioned papers only deal with natural images, for which the datasets available are usually larger than the medical datasets. The approach we suggest in this paper involves no generation of artificial images and requires no changes to the network architecture (and could indifferently be combined with one-stage or two-stage detectors), only a balanced way to sample images so that the distribution is uniform.

**Paper outline.** In what follows, we will start in Section 2 by explaining how the problem of imbalanced classes can be tackled using quadratic optimization, and how it can be solved in practice. Then, we will show in Section 3 how it was applied to a specific fetal anatomies detection problem, and the influence we observed on the training results, compared to several baselines.

## 2. Balanced sampling as quadratic programming

### 2.1. Introducing problem $(\mathcal{P}_\alpha)$ for sampling data

**Problem statement.** We now state the problem under its general form. Even if we aim at applying it to ultrasonic medical images, our approach is general and could be applied to any object detection problem.

Let $N$ be the number of images in the dataset, and $(X_1, X_2, \ldots, X_N)$ denote the collection of images. We also denote by $C$ the number of classes present in the dataset, and we consider that for any given image $X_i$ (with $i \in [\![1, N]\!]$), any label can be present in the

image. More precisely, we denote by $E$ the matrix encoding the distribution of labels within the image collection: $E := (\varepsilon_{i,l})_{1 \leq i \leq N, 1 \leq l \leq C} \in \mathcal{M}_{N,C}(\mathbb{R})$ with $\varepsilon_{i,l} = 1$ if label $l$ is present in image $i$, 0 otherwise.

Consider now a probability vector $p = (p_1, \ldots, p_N) \in \mathbb{R}^N$ (i.e., $p_i \geq 0$ for all $i \in [\![1, N]\!]$ and $p_1 + \ldots + p_N = 1$). If we randomly pick images amongst the collection $(X_i)_{1 \leq i \leq N}$ under the probability distribution $p$, the expectation to observe the class $l$ is $f_l := \sum_{i=1}^{N} p_i \varepsilon_{i,l}/N$. In order to have a balanced dataset, we therefore aim at finding a discrete probability vector (when possible) such that all expectations are the same. Therefore, the problem of sampling the images in a balanced fashion writes as follows: find a vector $p \in \mathbb{R}^N$, such that $p_i \geq 0$ for all $i \in [\![1, N]\!]$ and $p_1 + \ldots + p_N = 1$, and $f_l = f_k$ for all $l, k \in [\![1, C]\!]$, where $f_l := \sum_{i=1}^{N} p_i \varepsilon_{i,l}/N$. In what follows we will, with a slight abuse in the notations, forget the normalizing factor $1/N$, and still denote by $f_l$ the quantity $\sum_{i=1}^{N} p_i \varepsilon_{i,l}$, or in other words, $f = E^T p$.

Note that it may not always be possible to find such a vector $p$. To circumvent this issue, we consider instead the following optimization problem:

$$\text{Minimize} \quad \tfrac{1}{2} \sum_{l=1}^{C} \sum_{k=1}^{C} (f_l - f_k)^2,$$
$$\text{subject to} \quad \left\{ \begin{array}{c} p_i \geq \alpha, \\ p_1 + \ldots + p_N = 1. \end{array} \right. \tag{$\mathcal{P}_\alpha$}$$

The term $p_i \geq \alpha$ aims at ensuring that every image has a minimal probability to be picked in the sampling process, i.e. that no image is left unseen during training. (Note also that in order to have at least one solution, $\alpha \leq 1/N$ is required.)

Before going further, we perform some algebra on the cost function in $(\mathcal{P}_\alpha)$:

$$\begin{aligned} \sum_{l=1}^{C} \sum_{k=1}^{C} (f_l - f_k)^2 &= \sum_{l=1}^{C} \sum_{k=1}^{C} f_l^2 + f_k^2 - 2f_l f_k, \\ &= 2C \sum_{l=1}^{C} f_l^2 - 2 \sum_{l=1}^{C} \sum_{k=1}^{C} f_l f_k, \\ &= 2C f^T f - 2 f^T J_C f, \end{aligned} \tag{1}$$

where $J_C$ denotes the matrix in $\mathcal{M}_C(\mathbb{R})$ where every element is equal to 1. Finally, we get that the cost writes $f^T (2C I_C - 2 J_C) f/2$ and we denote by $A'$ the matrix $2C I_C - 2 J_C \in \mathcal{M}_C(\mathbb{R})$ ($I_C$ is the identity matrix of size $C$). Introducing $A = E A' E^T \in \mathcal{M}_N(\mathbb{R})$, we get that problem $(\mathcal{P}_\alpha)$ writes

$$\text{Minimize} \quad \tfrac{1}{2} p^T A p,$$
$$\text{subject to} \quad \left\{ \begin{array}{c} p_i \geq \alpha, \\ p_1 + \ldots + p_N = 1, \end{array} \right. \tag{$\mathcal{P}_\alpha$}$$

which is a standard form for a quadratic optimization problem.

## 2.2. Enforcing uniqueness

The solution to problem $(\mathcal{P}_\alpha)$ is (in general) not unique (see Appendix A for more details). In order to enforce uniqueness of the solution, we add to the initial problem a regularization term $\lambda \|p\|^2/2$ where $\lambda \geq 0$ is a penalization parameter. This yields the following optimization problem, which admits a unique solution (as long as $\alpha \leq 1/N$),

Table 1: Summary of bounding boxes anatomies and sub-anatomies, and the number of occurences in the training dataset. CSP stands for Cavum Septum Pellucidum

| main anatomies | sub-anatomies | | | |
|---|---|---|---|---|
| head (579) | falx cerebri (523) | cerebellum (114) | thalamus (408) | CSP (284) |
| abdomen (327) | umbilical vein (239) | stomach (279) | spine (349) | heart (36) |
| femur (303) | | | | |

$$
\begin{aligned}
&\text{Minimize} && \tfrac{1}{2}p^T A p + \lambda \tfrac{\|p\|^2}{2}, \\
&\text{subject to} && \left\{ \begin{array}{c} p_i \geq \alpha, \\ p_1 + \ldots + p_N = 1. \end{array} \right.
\end{aligned} \qquad (\mathcal{P}_{\lambda,\alpha})
$$

where $\alpha$ and $\lambda$ will be two hyper parameters, that can be set by the user for training. We refer to Appendix A for mathematical results on ($\mathcal{P}_{\lambda,\alpha}$), as well as some details on how it can be efficiently solved in practice.

## 3. Application to fetal anatomy detection

### 3.1. Data

We apply the method to fetal anatomy detection. The dataset comprises 2D frames from ultrasonic acquisitions of the head, abdomen and upper leg of a fetus. The aim is to detect and localize in the images some predefined anatomies, that correspond to our classes.

**Training dataset.**  Our training dataset consists in 1237 2D images. The target anatomies can be split into two categories that we will call *main fetal anatomies* and *sub-anatomies*. They are summarized in Table 1. Here, sub-anatomies are defined as structures that are part of a main anatomy, *e.g.*, a cerebellum is *always* included in a head (see also Figure 1 for some sample examples).

Given the definition of labels ('main anatomies' and 'sub-anatomies'), any fetal anatomy dataset is intrinsically imbalanced. Indeed, sub-anatomies are not present in all frames where their corresponding main anatomy is present (*e.g.*, a stomach may be missing, even though we visualize the abdomen), whereas the corresponding main anatomy is necessarily present when a sub-anatomy is present (*e.g.*, if we visualize the stomach, then the abdomen must be visible too). This leads to datasets in which heads and abdomens are over represented compared to inner structures, as shown in Table 1.

Besides, a common strategy consisting in duplicating images containing the less represented anatomies (for instance in our case, all the images containing a heart or a cerebellum), would also lead to an over-representation of the corresponding 'main anatomies', which may in turn introduce a new bias in the dataset. This will be supported by the experiments presented in Subsection 3.3, where we will compare our strategy to oversampling.

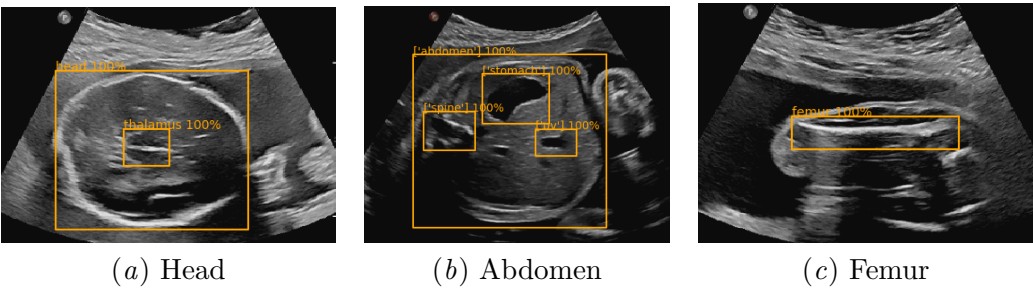

$(a)$ Head    $(b)$ Abdomen    $(c)$ Femur

Figure 1: Examples of dataset samples and bounding boxes

**Validation and test datasets.** We also use a small validation dataset (315 images) to monitor the loss and metrics during training and avoid over-fitting on the train dataset, and a final test dataset of 517 images on which all the statisitical evaluation is performed.

### 3.2. Setup

**Parameters.** In order to evaluate our method, we need to set the hyper-parameters $\alpha$ and $\lambda$, defined in Section 2.2. To do so, we simulated draws after solving the optimization problem for $\lambda \in [0, 1000]$ and $\alpha \in [0, 1/N]$.

We chose to set $\alpha = 0.5 \times 1/N$ as this ensures that each image is seen at worse twice fewer times than in the original training dataset distribution. We found it to be a good compromise between high data variety and class balance.

When $\lambda$ is too small, the anatomies are better balanced but with very few images over-represented in the training set. We found $\lambda = 500$ to offer a good compromise. High enough for regularization to become effective, but low enough to actually achieve a more balanced dataset in terms of anatomy distribution, as illustrated on Figure 2.

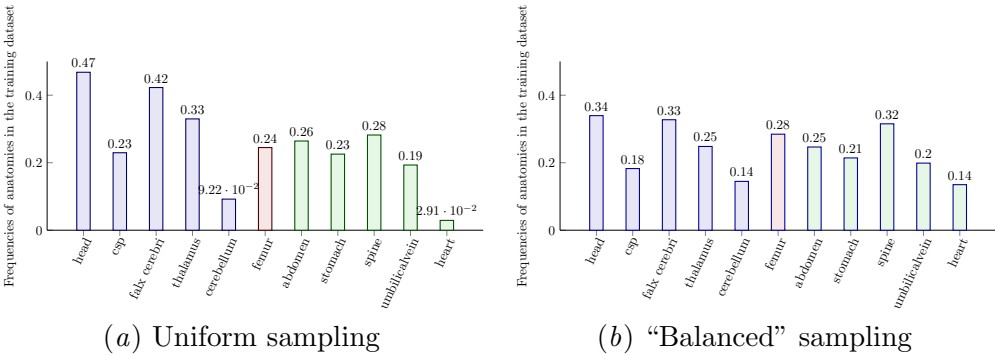

$(a)$ Uniform sampling    $(b)$ "Balanced" sampling

Figure 2: Impact of our sampling method (with $\alpha = 0.5 \times 1/N$ and $\lambda = 500$) on the frequencies of the different anatomies in the training dataset, compared to uniform sampling. Colors indicate anatomies that share the same main anatomies.

**Model.** One of the state-of-the-art deep learning models for object detection is the YOLO model (Redmon et al., 2016; Redmon and Farhadi, 2016). Because of its high speed, and in view of real-time usage, this is the baseline model that we use. The network's layers are initialized using weights pretrained on the VOC 2007 dataset (Everingham et al., 2015). We start from a learning rate of $1e-5$, then increase it by a factor 10 after 50 epochs, and then decrease it by a factor 5 regularly during training (approx. every 500 epochs). All networks are trained using Adam (Kingma and Ba, 2014) algorithm, and various data augmentation operations (scaling, translation, rotation, gaussian noise, flips) are also applied.

### 3.3. Results

To evaluate the impact of balanced sampling, we use three different training methods:

- **Image generator with *uniform sampling***: the original training set is uniformly sampled. This will be our first baseline strategy.

- **Image generator with *oversampling***: our two minority classes, namely heart and cerebellum are duplicated in order to artificially produce more training images, with a factor 2 for cerebellum, and 5 for heart. The choice of those factors was decided to get a frequency comparable to the remaining classes. This will be our second baseline.

- **Image generator with *balanced sampling***: a different probability factor is applied to each image of the training set, in order to obtain a more balanced dataset. This factor is determined by finding a solution to $(\mathcal{P}_{\lambda,\alpha})$ with $\alpha = 0.5 \times 1/N$ and $\lambda = 500$.

3.3.1. EVALUATION METHOD.

We evaluate each of the trained models with the mean Average Precision (mAP) metric. Due to the inherent stochastic nature of neural network's training (even with a fixed weight initialization), we trained several models for each method (30 times with *uniform sampling*, 12 times with *oversampling* and 11 times with *balanced sampling*) aiming to obtain a statistically significant comparison of the various training methods. Due to long training times (around half a day to train one model, on a GTX 1080 Ti, with input image size $416 \times 416$), we make the choice of evaluating only one setup of $\alpha$ and $\lambda$ in order to be able to conduct a thorough statistical evaluation and comparison with the baselines.

In order to compute the mAP, we need to set an Intersection Over Union (IoU) threshold $\theta$ that will separate false positives (IoU $< \theta$) from true positives (IoU $\geq \theta$) bounding boxes detections. In what follows, we will use the values $\theta = 0.2$ and $\theta = 0.4$.

We used the Kolmogorov-Smirnov test (KS) (for which we give more details in Appendix B) to compare performance distributions from both methods.

3.3.2. DETAILED RESULTS AND STATISTICAL ANALYSIS.

In Table 2 and Table 3, we provide average mAP results over all our trained models with $\theta \in \{0.2, 0.4\}$, as well as their respective standard deviations. For instance, for $\theta = 0.2$, the KS test between uniform sampling and balanced sampling provides us with a statistic $D = 0.476$ and p-value $P = 0.035$. Our strategy improves the overall performance of the models, both when compared to uniform or oversampling.

Table 2: mAP based on IoU threshold $\theta \in \{0.2, 0.4\}$ with a focus on under-represented anatomies such as the heart and cerebellum. The average, std and best mAP values over all trained models are presented, as well as the p-value for the statistical test comparing the distribution of performances for *uniform sampling* and our *balanced sampling* strategy.

| | Uniform sampling | | | Bal. sampling (ours) | | | |
|---|---|---|---|---|---|---|---|
| | average | std | best | average | std | best | p-value |
| all anatomies | | | | | | | |
| mAP @ $\theta = 0.2$ | 58.97 | 3.96 | 63.4 | **62.35** | 1.53 | **64.3** | .035 |
| mAP @ $\theta = 0.4$ | 54.87 | 3.98 | **59.6** | **57.75** | 1.48 | 59.3 | .087 |
| Heart | | | | | | | |
| AP @ $\theta = 0.2$ | 1.03 | 3.08 | 7.7 | **6.95** | 7.34 | **21.2** | .038 |
| AP @ $\theta = 0.4$ | 0.5 | 2.31 | 7.7 | **5.7** | 5.8 | **15.4** | .031 |
| Cerebellum | | | | | | | |
| AP @ $\theta = 0.2$ | 4.84 | 12.01 | 33.8 | **17.13** | 9.91 | **36.3** | .0002 |
| AP @ $\theta = 0.4$ | 3.72 | 9.4 | 28.6 | **14.9** | 9.4 | **36.3** | .0001 |

**Balanced sampling vs. uniform sampling.** As displayed in Table 2, the improvement appears when looking at specific sub-anatomies, such as the heart or cerebellum, which are under represented in the original dataset. With uniform sampling, the trained models perform very poorly on these structures, whereas the AP performance is greatly improved using balanced sampling.

**Balanced sampling vs. oversampling.** Duplicating images of the minority classes has boosted the performance of the networks on those classes, compared to the uniform strategy. As displayed in Table 3, our strategy and oversampling perform similarly on under-represented anatomies. However, our method has a better overall performance, showing that it is efficient in boosting the performance on under-represented classes, while not deteriorating the performance on the remaining classes.

### 3.3.3. Discussion

It is also interesting to notice than even if the average mAP over all trained models is improved with *balanced sampling*, it is more limited when focusing on the best performing trained model. In fact, depending on the IoU-threshold that is set for the mAP evaluation, the best model can be obtained with the strategy consisting in picking images uniformly (which needs to be mitigated by recalling that we trained the uniform baseline strategy three times more than the two others).

We interpret the difference as the fact that a balanced training dataset reduces the stochastic impact of data feeding to the network. The lower std values obtained with *balanced sampling* support this interpretation. It tends to make the training more *robust* and *reproducible* and enables to reach the best level of performance in a more systematic

Table 3: Comparison between *oversampling* and our *balanced sampling* strategy.

| | oversampling | | | Bal. sampling (ours) | | | |
|---|---|---|---|---|---|---|---|
| | average | std | best | average | std | best | p-value |
| all anatomies | | | | | | | |
| mAP @ $\theta = 0.2$ | 56.8 | 3.5 | 62.3 | **62.35** | 1.53 | **64.3** | .002 |
| mAP @ $\theta = 0.4$ | 51.9 | 3.5 | 57.2 | **57.75** | 1.48 | **59.3** | .004 |
| Heart | | | | | | | |
| AP @ $\theta = 0.2$ | 3.4 | 4.5 | 15.3 | **6.95** | 7.34 | **21.2** | .86 |
| AP @ $\theta = 0.4$ | 0.0 | 0.0 | 0.0 | **5.7** | 5.8 | **15.4** | .04 |
| Cerebellum | | | | | | | |
| AP @ $\theta = 0.2$ | **19.1** | 12.0 | **44.1** | 17.13 | 9.91 | 36.3 | .89 |
| AP @ $\theta = 0.4$ | 14.6 | 10.7 | **37.2** | **14.9** | 9.4 | 36.3 | .85 |

way, and with less tries. We believe this can be of great interest in practice, given the time and resources required to train deep neural networks.

## 4. Conclusion

In this paper, we suggest a new approach to deal with imbalanced datasets for object detection problems. During training, images are sampled following a probability distribution that helps bring balance between various classes. This probability distribution is computed beforehand by solving some quadratic optimization problem. Besides, the method is systematic, and can be applied to potentially any object detection problem. However, it requires tuning of the two hyperparameters $\alpha$ and $\lambda$.

We also showed how this sampling strategy impacted the performance of models: underrepresented structures become better detected, while it does not deteriorate the performance of the network on other structures. In fact, the average mAP performance increased by around 3% compared to uniform sampling (while the standard deviation of the performance was reduced from $\approx 4$ to $\approx 1.5$), and by around 5.5% compared to oversampling (while the standard deviation was reduced from $\approx 3.5$ to $\approx 1.5$).

A natural perspective would be to apply the technique to other object detection challenges, for instance on the COCO dataset (for natural images) or other medical imaging datasets, and further evaluate what it brings to the models' performance. Another interesting perpective would be to extend our evaluation to two-stage detectors, and combine our method with existing methods for two-stage detectors, for instance OFB, as they would act at two different levels of the training pipeline: before constitution of the batch (ours), and at the level of ROI proposals by the region proposal network (OFB).

## Acknowledgments

Both authors would like to thank Cybèle Ciofolo-Veit and Laurence Rouet for their valuable insight into both the clinical application and the scientific contribution of this work.

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

## Appendix A. Mathematical properties of $(\mathcal{P}_\alpha)$ and $(\mathcal{P}_{\lambda,\alpha})$.

**Non-uniqueness of a solution for $(\mathcal{P}_\alpha)$.** Depending on the matrix $A$, the solutions to problem $(\mathcal{P}_\alpha)$ are not necessarily unique. Let us for instance consider a toy problem with 2 classes and 3 images, one of which contains an occurrence of the first class, the two others containing an occurrence of the second class. That is, the matrix $E$ encoding the distribution of classes in the dataset is:

$$E = \begin{pmatrix} 1 & 0 \\ 0 & 1 \\ 0 & 1 \end{pmatrix} \tag{2}$$

Then, any vector $p \in \mathbb{R}^3$ of the form $p = (0.5, 1/x, 0.5 - 1/x)$ with $\alpha \geq x \geq 2/(1 - 2\alpha)$ is a solution to the problem (indeed, $E^T p$ is in this case an eigenvector of the matrix $A'$, associated to the eigenvalue 0).

**Structure of the solution to $(\mathcal{P}_{\lambda,\alpha})$.** For all $\lambda > 0$ and $0 \leq \alpha \leq 1/N$, the following result holds on the structure of the solution to $(\mathcal{P}_{\lambda,\alpha})$:

**Proposition 1** *For $\lambda > 0$ and $0 \leq \alpha \leq 1/N$, let $p^{(\lambda,\alpha)}$ denote the (unique) solution to $(\mathcal{P}_{\lambda,\alpha})$. If $X_{i_1}$ and $X_{i_2}$ are two images containing the same objects, i.e., the $i_1$-th and $i_2$-th lines of the matrix $E$ are the same, then the $i_1$-th and $i_2$-th components of vector $p^{(\lambda,\alpha)}$ are the same, $p_{i_1}^{(\lambda,\alpha)} = p_{i_2}^{(\lambda,\alpha)}$.*

This result shows that our sampling technique is consistent for images containing exactly the same objects: if two (or more) images have the same object distribution (*i.e.*, same corresponding lines in the matrix $E$), they will be assigned the same sampling probability, ensuring that none is favored over the other.

**Proof** For $\lambda > 0$ and $\alpha \leq 1/N$, we denote by $p^{(\lambda,\alpha)}$ the solution to $(\mathcal{P}_{\lambda,\alpha})$. Let $i_1$ and $i_2$ be two indices such that the $i_1$-th and $i_2$-th lines of $E$ are the same. We also denote by $(L_1, L_2, \ldots, L_N)$ the lines of the matrix $E$, and we therefore have $L_{i_1} = L_{i_2}$.

We show by contradiction that it implies that $p_{i_1}^{(\lambda,\alpha)} = p_{i_2}^{(\lambda,\alpha)}$. Assuming that $p_{i_1}^{(\lambda,\alpha)} \neq p_{i_2}^{(\lambda,\alpha)}$, we define the vector $q$ by:

$$q_i = \begin{cases} p_i^{(\lambda,\alpha)} & \text{if } i \neq i_1 \text{ and } i \neq i_2, \\ \frac{p_{i_1}^{(\lambda,\alpha)} + p_{i_2}^{(\lambda,\alpha)}}{2} & \text{if } i = i_1 \text{ or } i = i_2. \end{cases} \tag{3}$$

First, it is obvious that $q_1 + \ldots + q_N = p_1^{(\lambda,\alpha)} + \ldots + p_N^{(\lambda,\alpha)} = 1$, and that $q_i \geq \alpha$ for all $i \in [\![1, N]\!]$.

Then, we verify that the probabiliy vector $q$ yields the same frequency distibution as $p^{(\lambda,\alpha)}$:

$$
\begin{aligned}
E^T p^{(\lambda,\alpha)} &= \sum_{i=1}^{N} p_i^{(\lambda,\alpha)} L_i^T \\
&= p_{i_1}^{(\lambda,\alpha)} L_{i_1}^T + p_{i_2}^{(\lambda,\alpha)} L_{i_2}^T + \sum_{i \neq i_1, i_2} p_i^{(\lambda,\alpha)} L_i^T \\
&= 2 \cdot \frac{p_{i_1}^{(\lambda,\alpha)} + p_{i_2}^{(\lambda,\alpha)}}{2} L_{i_1}^T + \sum_{i \neq i_1, i_2} q_i L_i^T \\
&= q_{i_1} L_{i_1}^T + q_{i_2} L_{i_2}^T + \sum_{i \neq i_1, i_2} q_i L_i^T \\
&= E^T q.
\end{aligned}
\tag{4}
$$

We deduce from the equality (4) that

$$
\begin{aligned}
q^T A q &= (E^T q)^T A' E^T q \\
&= (E^T p^{(\lambda,\alpha)})^T A' E^T p^{(\lambda,\alpha)} \\
&= p^{(\lambda,\alpha)T} A p^{(\lambda,\alpha)}.
\end{aligned}
\tag{5}
$$

Finally, we compute the euclidian norm of $q$:

$$
\begin{aligned}
\|q\|_2^2 &= \sum_{i=1}^{N} q_i^2 \\
&= q_{i_1}^2 + q_{i_1}^2 + \sum_{i \neq i1, i2} q_i^2 \\
&= \frac{\left(p_{i_1}^{(\lambda,\alpha)} + p_{i_2}^{(\lambda,\alpha)}\right)^2}{2} + \sum_{i \neq i1, i2} \left(p_i^{(\lambda,\alpha)}\right)^2 \\
&< \left(p_{i_1}^{(\lambda,\alpha)}\right)^2 + \left(p_{i_2}^{(\lambda,\alpha)}\right)^2 + \sum_{i \neq i1, i2} \left(p_i^{(\lambda,\alpha)}\right)^2 \\
&< \left\|p^{(\lambda,\alpha)}\right\|_2^2.
\end{aligned}
\tag{6}
$$

Therefore, combining Equation (5) and (6), we get the following inequality on the cost of the optimization problem:

$$
\frac{1}{2} q^T A q + \lambda \frac{\|q\|_2^2}{2} < \frac{1}{2} p^{(\lambda,\alpha)T} A p^{(\lambda,\alpha)} + \lambda \frac{\left\|p^{(\lambda,\alpha)}\right\|_2^2}{2},
\tag{7}
$$

which contradicts the optimality of the solution $p^{(\lambda,\alpha)}$. ∎

**Solving** $(\mathcal{P}_{\lambda,\alpha})$**.** The optimization problem $(\mathcal{P}_{\lambda,\alpha})$ is a quadratic optimization problem, with linear constraints. This class of problems admits a wide variety of solving methods, one of which is *interior-point methods*. For this paper, we used the Python package `CVXOPT` (Andersen et al., 2011) which offers a convenient setting to define any optimization problem under the form

$$\begin{aligned}
\text{Minimize} \quad & \tfrac{1}{2}x^T Q x + q^T x, \\
\text{s.t.} \quad & \begin{cases} Gx \preceq h, \\ Ax = b. \end{cases}
\end{aligned} \tag{8}$$

Let us also mention the interior point optimizer `IPOPT` (Wächter and Biegler, 2006) (for general non-linear programming), which offers an interface with many programming languages, and can also be coupled with the modeling language `AMPL` (Fourer et al., 1993).

## Appendix B. Kolmogorov-Smirnov test

In this section, we make a general comment on our methodology to statistically evaluate the performance of two models. Assume that we want to compare the performance of a "new" model with a baseline model. The "new" model is trained $n$ times, with performance $x = (x_1, \ldots, x_n)$ following an (unknown) distribution $F$, and the baseline model is trained $m$ times, with performance $y = (y_1, \ldots, y_m)$ following a distribution $G$.

First, the empirical distribution functions need to be defined:

$$F_n(u) = \frac{1}{n} \sum_{i=1}^{n} \mathbb{1}_{]-\infty, u]}(x_i),$$

$$G_m(u) = \frac{1}{m} \sum_{j=1}^{m} \mathbb{1}_{]-\infty, u]}(y_j).$$

The Kolmogorov statistic is then defined as

$$D_{n,m} = \sup_{u \in \mathbb{R}} |F_n(u) - G_m(u)|$$

We consider the null-hypothesis $\mathcal{H}_0$ that the samples $x$ and $y$ follow the same distribution, *i.e.*, $F = G$. Given an observed value $d$ of the Kolmogorov statistic, and in order to reject (or not) the null-hypothesis $\mathcal{H}_0$, we aim at computing the p-value of the statistical test:

$$P = \mathbb{P}(D_{n,m} \geq d \mid F = G). \tag{9}$$

If the probability (9) is below a threshold $\alpha$, we reject $\mathcal{H}_0$ at level $\alpha$. In python, this statistical test is implemented with the function `ks_2samp`, within the package `scipy.stats`.

