# OpenReview forum: "Balanced sampling for an object detection problem - application to fetal anatomies detection"
_MIDL.io/2021/Conference — MIDL 2021_

### Official Review · AnonReviewer3 · 2021-03-05

**Confidence:** 3
**Preliminary Rating:** 3
**Recommendation:** Oral, Poster
**Final Rating:** 3

**Summary:**

In this paper, the authors address the problem of class imbalance in the context of object detection in medical images.
To tackle this problem, they propose a method to apply a balanced sampling of the different classes by calculating a probability of sampling distribution over the images in a pretraining step.
They validate this method on a fetal anatomies detection dataset using a Yolo detection model and obtain better performances compared to a uniform sampling strategy and a naive oversampling.

**Strengths:**

The paper is very well structured and written. I enjoyed reading it.

The research problem is stated clearly, which is really appreciated. The introduction is clear. The related work section could be more detailed but still succeeds in positioning the work properly.

The presentation of the proposed solution is well written. I appreciate the balance between textual explanations and mathematical formalism, leaving advanced details for the appendixes.
The conversion of the sums into matrices in equation (1) builds bridges between the theory and the implementation.

In 3.2.1, the choice of the hyperparameters is well detailed.

In 3.3.2, the statistical analysis of the results, presenting not only the best results but also the mean and standard is very good and something I would like to see more often in the papers.

The conclusion in 4 is well written, presenting not only positive points of the method (increased performance VS baseline) but also negatives (increased complexity due to the introduction of new hyperparameters)

**Weaknesses:**

What could be improved is mainly the experimental setup.
My main concerns are:
1) Why did you choose Yolo (and not Faster-RCNN for example which yields better performance)? And why only Yolo? Testing the method on different models would strengthen the paper a lot.
2) Also, validating the approach on other datasets would make the paper much stronger.
3) The Yolo model used is pre-trained on VOC and fine-tuned on the fetal anomalies detection dataset. Could you detail more how the fine-tuning is done? For example, if any part of the model is frozen or how you handle the learning rate.
Minor points:
4) In table 2, how were the thresholds chosen? And why only 2?
5) In the discussion section, I am missing a word about complexity VS performance. The method increases the performance on the final task but increases the complexity of the training. I am wondering if there is a way to quantify this increase of complexity, for example by measuring the extra time or extra computation needed.

**Deanonymize Review:**

no

**Detailed Comments:**

Nothing to add to my previous feedback.

**Final Rating Justification:**

The paper is in general of good quality. The authors did a good work in the rebuttal and answered my questions/concerns.
The main weakness I see is the limited validation (only on one dataset). I do not think applying the method on natural images is relevant for this paper/conference but I would have liked to see results on another medical imaging dataset. That's why I am maintaining my initial rating.

**Justification Of The Preliminary Rating:**

The results presented are limited (only one model and only one dataset). However, the idea is interesting, the method well presented and in general, the paper is of good quality. I consider it in this state as already enough for acceptance.

**Paper Type:**

methodological development

**Questions To Address In The Rebuttal:**

Any answer to my previous feedback would be helpful.


**Special Issue:**

yes

---

> ### Author Response · Authors · 2021-03-17
> **Answer to reviewer 3**
>
> Thank you for your time reading our manuscript.
>
> ### Model choice
> The choice of the YOLO architecture was motivated by a software requirement constraint to have an algorithm able to run real-time (With our system specification, YOLO is able to process twice as many images per second than Faster RCNN).
> However, we agree that it could be useful to combine our method with Faster RCNN, as other research teams may not have this real-time constraint, and we listed as a perspective to combine our method with two stage detectors in general, and the existing methods that were designed for two-stage detectors.
>
> ### Dataset
> We answered this comment in the common answer.
>
> ### Finetuning
> No freezing of the layers is performed. As for the learning rate, we start from a learning rate of 1e-5, then increase it by a factor 10 after 50 epochs, and then decrease it by a factor 5 regularly during training (approx. every 500 epochs). *We added a comment in the revised version.*
>
> ### Thresholds in Table 2
> We also performed an evaluation with 0.6 as a threshold, which confirmed the rest of the results. However, we noticed that in practice, 0.6 was a too restrictive threshold for our intended application, so we decided not to include it.
>
> ### Complexity vs. performance
> All the increase in complexity happens before the training, when computing the probability vector $p$ (and therefore choosing the two hyperparameters $\alpha$ and $\lambda$ - which is done very efficiently (less than one second) as QP problems have a wide variety of efficient solvers.
> Then, the training times and computational costs are the same than with other baselines.

---

### Official Review · AnonReviewer4 · 2021-03-08

**Confidence:** 4
**Preliminary Rating:** 1
**Final Rating:** 2

**Summary:**

The authors propose a novel approach to overcome foreground-foreground class imbalance in object detection. They compute a probability vector, encoding the probability of each image to be fed into the network for training. Their method uses quadratic programming to compute the probability vector. A regularization term is introduced into this to enforce a unique solution. This ensures that less frequent classes are seen more often during training in order to aid the network in generalization. The method is mathematically proven and applied to detect fetal anatomies from 2D ultrasound images.

**Strengths:**

The paper is generally well written and contains thorough proof of their method.
Their method performs significantly better than baselines and requires only minimal overhead.
They train their method multiple times in order to calculate standard deviation and p-values stating the significance of their results.


**Weaknesses:**

The baselines are very weak!
A follow up paper of one of their main references proposes a method regarding foreground-foreground class imbalance in object detection. This is not mentioned in the paper and would make for a great comparison.[1]

The relevance for the medical context could be discussed a bit more. Just using a medical dataset is not enough.
They evaluate only on one dataset.
Figure 1 & 2 are not mentioned in the papers text.


1.	Oksuz, K., Cam, B. C., Akbas, E., & Kalkan, S. (2020). Generating positive bounding boxes for balanced training of object detectors. In Proceedings of the IEEE/CVF Winter Conference on Applications of Computer Vision (pp. 894-903).


**Deanonymize Review:**

no

**Detailed Comments:**

Figure 1: what does the percentage tell us? e.g. [“spin”] 100%
It is not explained what the different colors in Figure 2 mean.
The figures could be placed better in the paper. Page 6 does not have another figure or table.
A figure representing the qualitative difference object detection would have been quite interesting.
The conclusion should be rewritten to end on a more upbeat note.

**Final Rating Justification:**

Though I acknowledge that not all research facilities have the capacity to run evaluation on multiple (or bigger) datasets, but conducting experiments only on one dataset exhibits serious risk of bias. As a result, this paper is lacking the experimental data to back the claims. This could be achieved either by comparing efficiency between other SOTA foreground-foreground class imbalance methods or by further comparisons on other datasets, even toy-like datasets.

**Justification Of The Preliminary Rating:**

The method is rigorously proven, but their experimental section lacks to support its usefulness. They further missed to compare to a recently proposed method (2020) for foreground-foreground class imbalance and an evaluation on more than one dataset should be conducted as their method tackles a class imbalance problem, which may greatly vary across different datasets.

**Paper Type:**

methodological development

**Questions To Address In The Rebuttal:**

How does this approach compare to other foreground-foreground class imbalance methods like Oksuz et al.[1] ?
What ranges of hyperparameters for alpha and lambda have been tested?
Have you been testing it on a COCO dataset or other medical datasets?


**Special Issue:**

no

---

> ### Author Response · Authors · 2021-03-17
> **Answer to reviewer 4**
>
> Thank you for your time reading our manuscript.
>
> ### Other foreground-foreground methods
> We find that the two most relevant existing papers to our work are [1] and [3]. However, we see some differences with what we suggest:
>
> #### In [1]:
> The method presented in this paper is designed to be integrated in two stage detectors: the authors suggest a new way to generate positive bounding boxes that can be used when training a network between the region proposal network (RPN) and the detector.
> In Section 6.1, the authors present the Online Foreground balancing method (OFB), aiming at making the various classes balanced within a batch. We see two main differences with the approach that we suggest:
>
> * First, our choice of the YOLO architecture was motivated by its ability to run real time, and was an important software specification for us. YOLO is a one-stage detector, and therefore, the predictions are not made from a set of bounding boxes from the RPN.
> Therefore, it is not obvious how to incorporate the suggested method in [1] in such an architecture. The authors of [1] state that their approach is more suitable for the two-stage methods, and their experiments are conducted using a Faster RCNN.
> For one-stage networks, they only state that "an additional module is required to employ [the] generator", but it is unclear to us how it could be combined with the YOLO architecture.
>
> * OFB makes sure that classes are evenly balanced within a batch, after the batch has been created, whereas our method intervenes before constitution of the batch.
> We can see this as an important difference, especially when some classes are heavily under-represented, and the batch size is small: in our case, hearts are present in less than 3% of the images. Therefore, with a batch size of 4, the probability to have an occurrence of a heart in a given batch is approx. 11% if images are sampled randomly, and hearts will remain less frequently seen by the network, even if OFB is applied a posteriori.
>
> * However, we find that a very interesting perspective could be to combine the OFB method of [1] and our balanced sampling strategy, as it acts at two different levels of the pipeline: before constitution of the batch (ours), and at the level of region proposals (OFB).
>
> _We added this discussion to the paper._
>
> #### In [3]:
> In [3], the authors investigate fine-tuning of a model on a dataset with \emph{long-tail distribution} and 200 classes (which is also a difference with the 11-class dataset that we present). They show that it is better to have (pseudo-)uniform number of samples per class, but sampling is done at bounding box level, before training the detector to classify each region, which is not easily generalizable to one-stage detectors .
>
> *We updated the literature survey part in the introduction with this discussion, and the conclusion section, to better position our work in the current state-of-the-art methods.*
>
> ### Choice of hyperparameters
>
> In order to set $\lambda$ and $\alpha$, we simulated draws after solving the optimization problem for $\lambda \in \left[0, 1000\right]$ and $\alpha \in \left[ 0.1 \times 1 / N, 1/ N \right]$.  *We added this precision to the paper.*
>
> We found $\lambda = 500$, $\alpha = 0.5 \times 1 / N$ to offer a good compromise for regularization to become effective, and at the same time actually achieve a more balanced dataset.
>
> ### Dataset
> We answered this comment in the common answer.
>
> ### Others:
> * Figure 1 & 2 are now again referenced in the text.
> * Colors indicate anatomies that share the same main anatomies, we added it to the caption. (blue for head anatomies, red for femur, and green for abdominal anatomies)
> * The percentage was used to display the score given to a class by the network. When we display a GT image, as in the case in Figure 1, we assign it the score 1, or 100%.
>
> [1] Generating positive bounding boxes for balanced training of object detectors, Oksuz et al, 2020
>
> [2] Imbalance Problems in Object Detection: A Review, Oksuz et al, 2019
>
> [3] Factors in finetuningdeep model for object detection with long-tail distribution, Ouyang et al, 2016

---

### Official Review · AnonReviewer1 · 2021-03-08

**Confidence:** 3
**Preliminary Rating:** 2
**Final Rating:** 3

**Summary:**

The authors present a method for performing balanced sampling of an imbalanced multi-class classification problem. The method relies on a quadratic minimization framing of the class probabilities.

The class balancing method is tested on a fetal ultrasound multi-class problem against uniform sampling and oversampling two minority classes. In repeated studies the proposed oversampling method performed better than the two naive baselines (measured by mean average precision).

**Strengths:**

The paper tackles an important problem of training machine learning models on imbalanced dataset. A problem that is prevalent in medical imaging settings. The paper is mostly well-presented.

The results demonstrate improved results over the two other methods compared against, showing potential for real-world use.

**Weaknesses:**

The major weakness of this paper is the lack of context for related work in this area. This manifests in two ways: both the introduction/discussion and experiments.

The imbalanced dataset problem is a well-studied problem with many existing rebalancing methods available. The paper mentions only two foreground-foreground papers, and does not compare the results with either of those existing methods.

A fairly common method is to oversample methods inversely proportional to its frequency, yet this is not used as a baseline. The paper would be significantly improved with more acknowledgement of the context of balancing methods and improved comparisons in the experiments. At current, only heuristic oversampling of two minority classes, and uniform sampling is compared against the proposed methods. These are decent positive controls, but they do not place the work in the context of existing work.

Note I am not setting a criterion that they must outperform all previous methods, but the relative performance to alternative ideas is important.

**Deanonymize Review:**

no

**Detailed Comments:**

In abstract: "solving some quadratic optimization problem", using "some" here sounds like the problem is unknown. I think it would be clearer as "solving a quadratic optimization..."

Table 1: does not have column separators to distinguish between the two columns. Inconsistent use of singular and plural names (e.g. "thalami" and "heart" in the table should be either "thalamus" and "heart" or "thalami" and "hearts").

In 3.1.1 "a stomach is always included in the abdomen", should that be that an image of a stomach is also always an image of the abdomen? This statement is not very clear. It is inconsistent with the previous statement that "stomach may be missing in an image of the abdomen"). It cannot be always included and sometimes missing.

3.2.1 "at worse twice less" -> "at worst twice fewer times". How is "overrepresentation" measured in 3.2.1?

3.2.2. "YOLO algorithm" -> "YOLO model"



**Final Rating Justification:**

I thank the reviewers for responding to my review.

They have improved my primary concern which was lack of context placement for this work in terms of previous work. They have added context to both the introduction and discussion.

However, there are still no comparisons with previous works (which is understandable given the short rebuttal period). I can therefore improve my rating to only 3 (weak accept).

**Justification Of The Preliminary Rating:**

I give this a weak reject, but it can be relatively trivially changed. With improved discussion and comparisons with existing (non-naive) methods I can see this being a valuable addition to the field. Specifically, the points mentioned in the "questions to be addressed".

**Paper Type:**

both

**Questions To Address In The Rebuttal:**

1. How does the method presented here differ from proportional oversampling?
2. How does it differ from some of the foreground-foreground methods presented in Oksuz et al. 2019 (ref in paper)? In particular, what are the strengths and weaknesses of this approach?
3. How would existing non-naive balancing methods perform on this task?

**Special Issue:**

no

---

> ### Author Response · Authors · 2021-03-17
> **Answer to reviewer 1**
>
> Thank you for your time reading our manuscript.
>
> ### Proportional oversampling
>
> Proportional oversampling does not apply in our case, as the presence of the various fetal anatomies is not independent: as several objects are present within each image, there is no way to decide which oversampling factor should be applied to an image as long as several objects are present in the image.
>
> This can even be problematic when occurrences of majority and minority classes are jointly present in the image: if such an image is consistently over-sampled, it may increase the frequency of the majority class; if it is consistently under-sampled, it will decrease the frequency of what is already a minority class.
>
> The experiment we conducted by oversampling the minority classes (with a factor 2 for cerebellum and a factor 5 for heart) is actually the best way we found to get close to what proportional oversampling would look like. We removed the term 'naive', as it may be misleading about the complexity of the task.
>
> Let us here also give an explanation on what was noted as unclear in paragraph 3.1.1, as it is closely linked to this question: We meant that if a stomach is visible in the image, then the abdomen also has to be visible (for anatomical reasons). However, there exists abdominal views where the stomach is missing (e.g., if a clinician is imaging the heart, or the bladder, or simply that the stomach is not visible).
> This is what we meant by "stomach may be missing in an image of the abdomen".  *We changed the formulation for better clarity*
>
> ### Other foreground-foreground methods
> We find that the two most relevant existing papers to our work are [1] and [3]. However, we see some differences with what we suggest:
>
> #### In [1]:
> The method presented in this paper is designed to be integrated in two stage detectors: the authors suggest a new way to generate positive bounding boxes that can be used when training a network between the region proposal network (RPN) and the detector.
> In Section 6.1, the authors present the Online Foreground balancing method (OFB), aiming at making the various classes balanced within a batch. We see two main differences with the approach that we suggest:
>
> * First, our choice of the YOLO architecture was motivated by its ability to run real time, and was an important software specification for us. YOLO is a one-stage detector, and therefore, the predictions are not made from a set of bounding boxes from the RPN.
> Therefore, it is not obvious how to incorporate the suggested method in [1] in such an architecture. The authors of [1] state that their approach is more suitable for the two-stage methods, and their experiments are conducted using a Faster RCNN.
> For one-stage networks, they only state that "an additional module is required to employ [the] generator", but it is unclear to us how it could be combined with the YOLO architecture.
>
> * OFB makes sure that classes are evenly balanced within a batch, after the batch has been created, whereas our method intervenes before constitution of the batch.
> We can see this as an important difference, especially when some classes are heavily under-represented, and the batch size is small: in our case, hearts are present in less than 3% of the images. Therefore, with a batch size of 4, the probability to have an occurrence of a heart in a given batch is approx. 11% if images are sampled randomly, and hearts will remain less frequently seen by the network, even if OFB is applied a posteriori.
>
> * However, we find that a very interesting perspective could be to combine the OFB method of [1] and our balanced sampling strategy, as it acts at two different levels of the pipeline: before constitution of the batch (ours), and at the level of region proposals (OFB).
>
> _We added this discussion to the paper._
>
> #### In [3]:
> In [3], the authors investigate fine-tuning of a model on a dataset with \emph{long-tail distribution} and 200 classes (which is also a difference with the 11-class dataset that we present). They show that it is better to have (pseudo-)uniform number of samples per class, but sampling is done at bounding box level, before training the detector to classify each region, which is not easily generalizable to one-stage detectors .
>
> _We updated the literature survey part in the introduction, and the conclusion section, to better position our work in the current state-of-the-art methods._
>
> We also corrected the various typos in the revised version of the manuscript as suggested.
>
> [1] Generating positive bounding boxes for balanced training of object detectors, Oksuz et al, 2020
>
> [2] Imbalance Problems in Object Detection: A Review, Oksuz et al, 2019
>
> [3] Factors in finetuningdeep model for object detection with long-tail distribution, Ouyang et al, 2016

---

### Official Review · AnonReviewer2 · 2021-03-09

**Confidence:** 5
**Preliminary Rating:** 4
**Recommendation:** Oral

**Summary:**

This work tackled the class-imbalance problem in object detection. Even though defect standard object-detection methods base on deep learning and their performance depend on training data, a balance between different classes is often hard to achieve and results in so-called foreground-foreground class imbalance. Therefore, we usually have imbalanced dataset for the training of object detector. The authors proposed balancing sampling as a preprocessing of each batch to mitigate the imbalance in training data. In this balancing method, how to sample from each class images is given by the solution of quadratic optimisation problem.

**Strengths:**

Interesting work. The imbalance problem is essential problem for machine learning and its applications. Even though many works focus on foreground-background class imbalance problem such that polyp detection—two classes of polyp and background—, this work focuses on foreground-foreground class imbalance, where multi classes exist.

The proposed method sounds technically. The proposed method randomly sampled images from imbalanced dataset following the probability distribution, which is computed before the training by solving a quadratic optimisation problem. This optimisation finds probability distribution that gives equal expectation of class frequency for all classes.

The authors demonstrated the validity of the proposed method by comparing the detection performances among the proposed, and the classical two subsampling- and oversampling-based method in the application to fetal anatomy detection. In this comparison, the proposed method improved the mean average precision for small-number-image classes more than the classical methods. As the results, the proposed method achieved the best performance over all anatomies among the three methods.

**Weaknesses:**

Setting of data splitting in experiments is unclear. For fair evaluation, dataset should be split into training, validation and test data without duplication of patients. Without this no-duplication splitting, we cannot evaluate the generalisation ability of the trained model. The presented experiments look only the evaluation in training data. I'm interested in the generalisation ability of the model trained with the proposed method.

Only one dataset is adopted in experiments. Evaluations with a few dataset are welcome for the demonstration of the validity of the proposed method.

Survey also looks limited.

**Deanonymize Review:**

no

**Detailed Comments:**

In the first term of the third line of Eq. (1), the center dot has to be omitted. It looks typo.

**Justification Of The Preliminary Rating:**

The proposed sampling method looks unique compared with other researches about imbalance problem. The paper is solidly written; the authors presented full mathematical definitions. Even though the experimental setting is limited, the validity of proposed method is give in their experimental evaluations.

There is no optimal solution to the imbalanced problem in machine learning field so far, and no powerful state-of-the-art method in recent works. The comparison with the classical approaches might be acceptable as a fair evaluation.

**Paper Type:**

both

**Special Issue:**

yes

---

> ### Author Response · Authors · 2021-03-17
> **Answer to reviewer 2**
>
> Thank you for your time when reading our manuscript.
>
> ### Data splitting
>
> The 1237 images dataset presented in the paper (section 3.1) is indeed the training dataset (on which our sampling technique is applied).
> However, even though it was missing from the original version of the paper:
> - We use a small validation dataset (315 images) to monitor the loss & metrics during training and avoid over-fitting on the train dataset;
> - All the statistical evaluation is performed on a test dataset, composed of 517 images that have therefore never been seen during training, to measure the generalization ability of the networks.
> This precision has been added to the paper.
>
> ### Dataset
>
> We answered this comment in the common answer.
>
> ### Survey
>
> As pointed out by reviewers 1 & 4, we included some missing references to the literature survey section (that we made explicit in our answer to reviewers 1 & 4), as well as a more thorough discussion to compare our method, and its application range, to existing methods.

---

### Author Response · Authors · 2021-03-17
**Common answer to all reviewers**

We warmly thank all reviewers for their time and feedback on our manuscript, especially for pointing out the weaknesses of our work. We will try in the following answers to address all concerns as well as possible.

We updated a revised version of the paper that includes our changes.

As 3 reviewers out of 4 mentioned the use of other datasets to make the paper stronger, we answer this comment here.
We agree that a very natural perspective for this work would be to apply the method to bigger natural images datasets (such as COCO, or PASCAL VOC).
Unfortunately, our organization focuses on medical applications, which is why we worked on the fetal anatomy dataset. For now, we have not had the time and resources to work on natural image datasets, especially given the amount of experiments required to conduct the statistical analysis we performed. Hence, we list this as a perspective for this work.
This is also the reason why we chose MIDL, a medically focused conference, to present this work.

---

### Meta-Review · Area_Chairs · 2021-03-29

**Recommendation:** Accept (Oral & Special Issue Candidate)

**Metareview:**

The paper receives overall positive comments from four knowledgeable and independent reviewers. They all like the novelty of the proposed work in addressing imbalanced sampling --- one of the most serious issues in medical image analysis. However, they also share the common concern, that is, lack of sufficient validation. Currently, only one dataset is used.  The authors also provide a rebuttal about this. While I agree with the argument they present (time and resource constraints, medical imaging focus, etc.), it is not difficult to find another medical imaging dataset to test their idea. Therefore, I strongly encourage the authors to conduct such an additional experiment to make their final version much stronger.

**Paper Type:**

methodological development

---

### Decision · Program_Chairs · 2021-03-31

**Decision:**

Accept

**Comment:**

Congratulations your paper has been selected as a long oral.